# Combined Effects of Surface Roughness, Solubility Parameters, and Hydrophilicity on Biofouling of Reverse Osmosis Membranes

**DOI:** 10.3390/membranes14110235

**Published:** 2024-11-08

**Authors:** Neveen AlQasas, Daniel Johnson

**Affiliations:** Water Research Center (WRC), Division of Engineering, New York University Abu Dhabi, Abu Dhabi P.O. Box 129188, United Arab Emirates

**Keywords:** Hansen solubility parameter, Hansen solubility distance, membrane roughness, membrane hydrophilicity, membrane fouling, biofouling

## Abstract

The fouling of protein on the surface of reverse osmosis (RO) membranes is a surface phenomenon strongly dependent on the physical and chemical characteristics of both the membrane surface and the foulant molecule. Much of the focus on fouling mitigation is on the synthesis of more hydrophilic membrane materials. However, hydrophilicity is only one of several factors affecting foulant attachment. A more systematic and rationalized methodology is needed to screen the membrane materials for the synthesis of fouling-resistant materials, which will ensure the prevention of the accumulation of foulants on the membrane surfaces, avoiding the trial and error methodology used in most membrane synthesis in the literature. If a clear correlation is found between various membrane surface properties, in combination or singly, and the amount of fouling, this will facilitate the establishment of a systematic strategy of screening materials and enhance the selection of membrane materials and therefore will reflect on the efficiency of the membrane process. In this work, eight commercial reverse osmosis membranes were tested for bovine serum albumin (BSA) protein fouling. The work here focused on three surface membrane properties: the surface roughness, the water contact angle (hydrophilicity), and finally the Hansen solubility parameter (HSP) distance between the foulant understudy (BSA protein) and the membrane surface. The HSP distance was investigated as it represented the affinities of materials to each other, and therefore, it was believed to have an important contribution to the tendency of foulant to stick to the surface of the membrane. The results showed that the surface roughness and the HSP distance contributed to membrane fouling more than the hydrophilicity. We recommend taking into account the HSP distance between the membrane material and foulants when selecting membrane materials.

## 1. Introduction

Membrane fouling, as well as its reduction, has been a subject of many academic and industrial studies since the early 1960s when the commercialization of membrane separation processes emerged. Fouling is the deposition of retained particles, colloids, macromolecules, salts, etc., at the membrane surface or inside the pore at the pore wall. Fouling eventually results in reduced water flux across the membrane. Fouling can be divided into two portions: reversible or irreversible, with the reversible fouling being the proportion of flux that can be recovered after membrane cleaning. Fouling is caused by the interaction between the membrane surface and the foulants, which can include inorganic, organic, and biological substances in many different forms [1].

When using naturally sourced waters or wastewater as a feed, biofouling is one of the most problematic sources of fouling. Biofouling is initiated by the conditioning of the membrane surface with dissolved organics and biological macromolecules, followed by microbial cell adhesion. Cells then excrete a mix of extra-cellular polymeric substances (EPSs) to form a biofilm, which greatly increases the flow resistance across the membrane [2]. Chemical disinfection can be an effective tool in minimizing biofilm formation. However, it is important to strike a balance between maintaining a biofilm-free environment and preserving the integrity of the membrane material, as well as minimizing the cost and environmental impacts associated with routine biocide use. Some common cleaning chemicals, such as sodium hypochlorite, can result in a degradation in the membrane material’s lifetime [1]. EPS is mainly composed of proteins, lipoproteins, lipids, polysaccharides, glycoproteins, and nucleic acids [3,4]. A chemical cleaning strategy can reduce the thickness of the biofilm; however, after a certain stage of biofilm development, chemical cleaning strategies cannot completely remove it, and the biofouling becomes partially irreversible [5]. Pressure-driven membrane processes such as reverse osmosis (RO), nanofiltration (NF), ultrafiltration (UF), and microfiltration (MF) can be severely affected by the formation of biofilms. Therefore, one of the most important factors to be considered in minimizing or eliminating the formation of fouling on the membrane surfaces is the membrane material selection. The membrane tendency to fouling is affected by various surface properties, such as morphology, hydrophilicity, and chemical groups that attach to the membrane [5]. The fouling mechanism varies depending on the foulant; therefore, studying each type of foulant separately is an essential step in studying the effect of various surface properties on fouling.

In this work we focus on protein fouling, using BSA as a model protein foulant molecule. Several studies have shown that the key irreversible foulants on polymeric membranes during biofouling are proteins [6,7,8,9]. The surface properties of the membrane determine the type of interaction between the specific foulant and the membrane surface. Most of those surface properties are related to each other [10,11,12,13,14,15,16]. For example, it is well known that the surface energy, adhesion, and adsorption of particles to surfaces are all related [13,17,18]. The chemical compositions of the surfaces can also affect the surface morphology and hydrophilicity of the membrane. While a large number of studies [17,19,20,21] have investigated how the different surface properties affect the strength of adhesion and adsorption of various biofoulants to different membrane surfaces and how those surface properties correlate with adhesion and adsorption, the picture is not yet entirely clear, due to the complicated interaction of the different relevant surface properties, the large number of foulants, and the different process operating parameters. If a clear correlation is established between the combined effects of surface properties and the adhesion and adsorption of biofoulants, a systematic approach can be made to the discovery and design of novel materials for membrane fabrication where biofouling is an issue.

A number of studies have shown that the hydrophilicity of the membrane surface plays an essential role in minimizing the protein fouling on to the surfaces [22,23,24,25,26]. However, various contradicting studies showed that hydrophobic surfaces showed less fouling [27,28,29]. Those contradicting findings are certainly related to the different types of foulants that are tested in various studies and to the type of water being treated. Likewise, very few studies have shown that membrane fouling directly correlates to the hydrophilicity/hydrophobicity of membranes across a wide range of water contact angle values [1,22]. Some have shown that hydrophilic membranes have higher permeation rates but not lower fouling or fouling reversibility [28]. However, even the water permeability of membranes does not always increase with the increase in hydrophilicity, as there exist a moderate hydrophilicity that causes the membrane to be adequately wetted by the feed solution at the target operating pressure drops [30]. The feed solution characteristics was found to have a pronounced effect on BSA fouling on RO membranes. For instance, the study by Ang and Elimelech [31] showed that the presence of alginate as a co-foulant with BSA enhanced the BSA fouling, and also, BSA fouling was enhanced at higher calcium concentrations of the treated water. Similar findings were reported by Zhao et al. [32]. Other surface properties, such as surface charge, surface roughness, morphology, and chemical structure, all play a major role in determining the number of foulant attachments to the various surfaces. Electrostatic interaction has been reported to be a key factor in controlling protein adsorption behavior on membrane surfaces [24,33], although charge shielding is likely to minimize this effect in even low-concentration electrolyte solutions.

The Hansen solubility parameters (HSPs) have been used in various applications for material screening and solvent selections [34,35,36]. It is also well known that the HSP is closely related to the surface energy and adhesion characteristics of any surface [17,37]. Various researchers have established correlations between surface energies, interfacial energies, adsorptions, and HSPs [13,14,18,37,38,39]. The HSP has been widely used in various applications [40,41,42,43,44,45,46,47,48,49,50,51,52] including optimizing solvent selection, improving polymer computability, enhancing pigment dispersion, etc. It has also been used in membrane technology to study the mass transfer, as in the work of Huang et al. [53]. Huang et al. combined theory and experiments to quantitatively analyze the relationship between the HSP and the change of membrane mass transfer channel using swelling as a bridge between the HSP and the SCR parameter. A new parameter, the swell-cavity ratio (SCR), which characterizes changes in mass transfer channels, was defined as the ratio of molecular weight cut-off (MWCO) before and after swelling and was used to construct a theoretical model with total HSP. Their model showed that the HSP distance between the membrane and solvents (which is a representation of the affinity between two molecules) is strongly related to the swelling degree of the membrane, which provides a convenient alternative theoretical parameter for the quantitative model [54,55,56,57]. Huang et al. [58] developed a versatile HSP-guided concept to design a high-performance mixed matrix membrane. Based on the HSP of the different components used for the fabrication of the mixed matrix membrane (MMM), Huang et al. enhanced the filler dispersion in the MMM structure and therefore produced an MMM that simultaneously exhibited improved permeability and selectivity for CO_2_/N_2_ separation. Azeem et al. [59] used HSPs to study the effect on the thermodynamics of membrane formation and found good correlation with experimental observation. The membrane morphology and performance were correlated with polymer and solvent variation from the consideration of Hansen solubility parameters. Shin et al. [60] determined the HSP of crosslinked polyamide (PA) membranes by quantifying the swelling degree of a molecular layer-by-layer assembly and a model PA nanofilm in various solvents via in situ atomic force microscopy. They incorporated the newly determined HSPs of the PA layer in combination with free volume and Flory–Huggins solution theories into the solution-diffusion model. This refined solution-diffusion model accurately predicted the permeance of 16 different solvents by capturing the characteristics of solvent–membrane affinity. The effect of HSP in the separation of different solvent mixtures or solvent–water mixtures has been reported in the literature by several workers as well [61,62,63,64]. HSP has also been used to study the performance of new green solvents intended to replace the conventional toxic solvents in membrane preparations [65,66].

Despite the use of HSPs in membrane fabrication and separation performance, not much use of the HSP theory had been utilized in screening materials used for the fabrication of superior anti-biofouling membrane materials. Very few studies have been conducted on the relationship between the HSP distance between the foulants and the membrane material and the rate of fouling occurring on the surface of the membrane. Fang et al. [18] showed that there exists a relationship between the HSP distance between polymeric films and BSA and the amount of BSA adsorbed at the surfaces of those polymeric films during static adsorption tests.

In this work, we present an investigation into the relationship between the HSP distance between BSA proteins and various commercial membrane surfaces and the amount of reversible and irreversible fouling when using a dead-end filtration setup. The membrane surfaces were characterized using several characterization methods, such as atomic force microscopy (AFM), scanning electron microscopy (SEM), Fourier transform infrared (FT-IR), and water contact angle. The HSPs of the surfaces of the commercial membranes were estimated via our previously reported method [67] from solvent contact angle data using a machine learning technique. The objective here was to investigate the existence of correlations between any or all of the tested surface properties and the fouling on the membrane surfaces and discuss the possibility of using such correlations in the material selection of membrane materials that will eventually minimize the biofouling attachment to the surfaces. In this work, the roughness, hydrophilicity, and affinity represented in HSP distance were considered to be studied in terms of the surface characteristics of the membranes.

### Theoretical Background of HSPs

The HSPs are physicochemical parameters that are widely used to estimate the interactions responsible for compatibility between materials. The cohesive energy (E) can be divided into three parts corresponding to atomic dispersion (E_d_), molecular dipolar interactions (E_p_), and hydrogen-bonding interactions (E_h_). Similarly, the total solubility parameter can be divided into three components corresponding to the abovementioned different types of molecular interactions: dispersion (δ_d_), polar (δ_p_), and hydrogen-bonding (δ_h_). Therefore, the HSP is defined according to the following equation, where δ_T_ is the Hildebrand solubility parameter [11]:(1)$$\delta_{t}^{2}=\delta_{d}^{2}+\delta_{p}^{2}+\delta_{h}^{2}$$

If two materials have known HSPs, it is possible to estimate their compatibility, miscibility, or dissolvability in each other using the HSP distance equation:(2)$$\mathrm{Ra}=\sqrt{{\left(\delta_{d1}-\delta_{d2}\right)}^{2}+{\left(\delta_{p1}-\delta_{p2}\right)}^{2}+{\left(\delta_{h1}-\delta_{h2}\right)}^{2}}$$
where Ra is the HPS distance between materials 1 and 2. The larger this amount, the less compatible those two materials and vice versa [68].

## 2. Materials and Methods

### 2.1. Commercial Membranes

Eight commercial reverse osmosis (RO) flat sheet membranes were used in this study. All the membranes were purchased from Sterlitech. Table 1 shows the names of the RO commercial membranes used in this study. Appendix A shows the detailed specifications of those commercial membranes from the manufacturer. All membranes were cut to a circular shape with a dimeter of 5 cm and soaked in deionized water for at least 5 days. To ease the use of the names of the commercial membranes in this manuscript, the number corresponding to the commercial membranes shown in Table 1 will be used frequently throughout the manuscript.

### 2.2. Biofoulant

Bovine serum albumin protein (BSA) was used as a model biofoulant. BSA was purchased from Sigma Aldrich (Burlington, VT, USA). A stock solution of BSA was freshly prepared to the required concentration (200 ppm), by dissolving 50 mg of powder in 250 mL of 18 MΩ·cm deionized water (Milli-Q, Millipore, Darmstadt, Germany) followed by continuous mixing of the solution.

### 2.3. Reagents and Chemicals

All chemical solvents used in this work were purchased from Sigma Aldrich. Ethylene glycol (EG), formamide (F), and dimethyl formamide (DMF) were used in the HSP analysis using contact angle measurements on the membranes’ surfaces. Solutions were made using high-purity water (18.2 MΩ s).

## 3. Material Characterization

### 3.1. Atomic Force Microscopy

The surface roughness of all the tested commercial membranes was measured before and after fouling using a Dimension Icon (Bruker, Billerica, MA, USA) atomic force microscope (AFM) in quantitative nanomechanical imaging mode using Scanasyst Air (Bruker) probes with a nominal tip radius of 10 nm and a spring constant of 400 mN/m. At least three imaging sites were selected arbitrarily and imaged with a scan size of 20 µm.

### 3.2. ATR-FTIR

The eight commercial membranes were analyzed for structural composition using a Thermo Scientific (Waltham, MA, USA) Nicolet IS5 FT-IR Spectrometer using an attenuated total reflection (ATR) module.

### 3.3. Scanning Electron Microscopy (SEM)

A Thermo/FEI Quanta 450 Field Emission SEM was used to measure the morphology of the membranes. The images were acquired using an Everhart–Thornley (ET) secondary electron detector with a 5 kV acceleration and a spot size of 2.

All samples were subjected to gold-coated SEM analysis using a Cresswell sputter unit using argon as the working gas, a 99.99% gold target, and a planetary rotation stage for uniform coverage. The gold deposition rate was calibrated via a reference film deposited on a microscope slide and measured via AFM to be 0.5 nm/s.

### 3.4. Water Contact Angle

Static advancing water contact angle measurements were recorded for each membrane studied. Measurements were performed using the sessile drop method with a Drop Shape Analyzer-DSA100S (Kruss Scientific, Hamburg, Germany). The droplet volume was set at 2 µL. The water contact angle was measured on up to 10 different locations on the selective layer surface of each membrane.

### 3.5. HSP Estimation of Commercial Membranes and BSA Protein

The HSPs of the surfaces of the membranes were all estimated using our recently proposed method [67]. This method utilized machine learning to predict the HSPs of the surfaces of different materials via simple contact angle measurements using three solvents only: F, EG, and DMF. The inputs to this model were the contact angle values of those three solvents with the membrane surface and their corresponding viscosity and air–liquid surface tension values. The output was the HSP distance between each solvent and the tested membrane. In order to estimate the HSP of each tested material, the Hansen solubility distance equation (Equation (2)) was solved for the three different solvents to estimate the HSP of the tested material. More details on this method can be found in our previous publication [67].

The HSP of BSA protein was estimated using UV-vis spectrometry ranking techniques with the use of HSPiP software (version 5.4.01) [69]. The detailed procedures are presented in our previous work [70]. Table 2 shows the HSP values of each of the commercial membranes and BSA protein as obtained in this study. The values of the HSP presented in Table 2 are the average values from at least three repeat runs. The average Ra between BSA and each membrane surface was calculated using Hansen’s HSP distance equation (Equation (2)) and is shown in Table 2 as well.

### 3.6. Dead-End Experimental Setup and Procedures

The commercial membranes shown in Table 1 were used in a dead-end filtration setup (Figure 1). Each filtration test for each membrane was repeated a minimum of three times. The following procedures were followed to conduct the filtration experiments:The membranes were washed and soaked in high-purity water for 4–5 days.Before starting the filtration experiment, the membrane under study was backwashed for about an hour to make sure that all preservatives were completely washed off. Backwashing was performed by inserting the membrane into the filtration setup, however, flipped (the active face facing the bottom side).After backwashing, the membrane was flipped in the dead-end filtration cell, deionized water was added to the filtration cell, and a pressure of 10 bar was applied for 30 min to allow the membrane to undergo compression.After compressing the membrane, the pressure was reduced to 8 bars (the experimental operating pressure) and was run for 30 min to allow the flow to stabilize before starting the experiment.Each experiment consisted of three runs: the first used high-purity water only as a feed; the second run used a BSA solution as a feed (200 ppm). The membrane was then backwashed to remove any reversable fouling, and after that the third run was performed again using high-purity water.For all three runs, the volume of permeate was recorded with time. Each run lasted for 2 h, and the stirrer speed was kept at 300 rpm.The steady state permeation flow rates were recorded after each run: J_1_, J_2_, and J_3_ for run 1, run 2, and run 3, respectively. In each case, the permeation rate was almost constant after approximately 30 min from the start of the experiment, and the steady state flow rate was considered as the average flow rate for the duration of the experiment after stabilization (the average flow rate for the last hour of experiment).The flux recovery ratio (FRR) was calculated according to the following relation:
(3)$$\mathrm{FRR}=\left(\frac{J_{3}}{J_{1}}\right)\times 100$$
where J_1_ is the flux (or flow rate) of the permeate of run 1 (pure water), and J_3_, is the flux/flow of the permeate of run 3 (pure water for membrane after backwashing).The reversable and irreversible fouling percentage (R_r_%, R_ir_%) were both calculated according to the following relations:
(4)$$R_{r}\%=\left(\frac{J_{3}-J_{2}}{J_{1}}\right)\times 100$$
(5)$$R_{\mathrm{ir}}\%=\left(\frac{J_{1}-J_{2}}{J_{1}}\right)\times 100$$The above procedures were repeated for all the 8 membranes at least three times, and the average FRR, R_r_%, R_ir_, and the therefore total fouling percentage R_t_% were calculated.

## 4. Results and Discussion

Figure 2 shows the average steady state permeation flow rates for the three previously described runs (runs 1, 2, and 3) for each of the eight commercial membranes. J_1_ is the average steady state permeation flow rate of the initial run with pure water as a feed, J_2_ is the average steady state permeation flow rate of the BSA solution as a feed, and J_3_ is the average steady state permeation flow rate of the pure water run after backwashing of the membrane. The highest permeation flow rate was observed for membrane XLE-PA (membrane 4), and the lowest permeation flow rate was observed for the SW30-XLE membrane (membrane 3). However, membrane 4 did not show the best flux recovery, as it suffered a 12% flux reduction after backwashing, as shown in Figure 3. For CR100 (membrane 8), although it did not show the highest permeation flow rate, its flux reduction percentage after the backwashing did not exceed 2.5%. UTC73-HA (membrane 1) suffered from the highest degree of fouling, with a reduction in the permeation flow rate of 41%.

### 4.1. Correlations Between the Surface HSP, Fouling Percentage, and Flux Recovery Ratio (FRR)

The filtration tests were performed as described in Section 3.6 on eight commercial membranes. The HSP distance (Ra) between each membrane surface and the BSA molecule was calculated based on Hansen’s equation (Equation (2)), and the average Ra values between BSA and each of the commercial membrane are shown in Table 2. The FRR, R_r_%, and R_ir_% for each experiment were calculated using Equations (3), (4), and (5), respectively. The filtration experiments were repeated at least three times, and therefore, all the results shown are for the average values.

Figure 4 shows the flux recovery ratio for the membranes as a function of the average HSP distance between BSA and the membrane surfaces. The overall trend shows an increase in the flux recovery ratio with the increase in the average Ra. The results are consistent with the HSP theory as the HSP distance between the BSA molecule and the surface of the membrane is high, indicating lower affinity; therefore, less fouling is expected, and this will result in higher flux recovery ratio and vice versa. The data in Figure 4 were fitted to a logarithmic equation with R^2^ = 0.596; if fitted to a straight-line relation, the R^2^ would be R^2^ = 0.57. The moderate correlation exhibited in Figure 4 indicates some likely relationship between the HSP distance and the fouling recovery rate.

Figure 5 shows the HSP distance versus the percentage of irreversible fouling on the surface of the membranes. Again, the straight-line correlation showed R^2^ = 0.57; however, the logarithmic relation showed a better R^2^ = 0.596. The overall trend showed lower irreversible fouling percentage with the higher HSP distance. The reversible fouling percentage R_r_% showed a poor correlation with the HSP distance (R^2^ did not exceed 0.05 for both logarithmic and straight-line relationship; figure not shown), indicating that the loosely adhered reversible fouling layer attachment was not related to the HSP distance between BSA and the membrane surface, suggesting it was foulant–foulant interactions only that governed the attachment of this layer. The more tightly bound irreversible fouling, however, did show correlation with the BSA–membrane HSP distance, suggesting that the physico-chemical interactions could be to some extent accounted for with HSPs.

As the individual HSPs were estimated for each membrane (Table 2), a high degree of correlation was found specifically relating the hydrogen bonding component, δ_H_, to Ra_avg_ between the membrane surface and BSA (Figure 6), showing that materials with higher values of δ_H_ had higher affinity to BSA molecules. The hydrogen bonding component of the BSA molecule as measured in this study (Table 2) was 17.5 Mpa^0.5^. Therefore, this may explain the tendency of BSA to attach to membranes with higher values of the δ_H_ parameter. This may indicate that the mechanism of attachment of the BSA molecule to the surface of the membranes is likely driven via formation of hydrogen bonds.

### 4.2. Correlations Between Surface Roughness, Fouling Percentage, and FRR

It has been previously reported [1,71,72,73] that the roughness of the surface of the membrane has a pronounced effect on the amount of fouling on the surface of the membrane. The surface roughness (R_q_) values of the eight commercial membranes represented were measured using AFM. Figure 7 and Figure 8 show the FRR and the R_ir_% as a function of R_q_, respectively. Figure 9 shows 3D images of the topography of the eight pristine membranes obtained by AFM. The SEM images of each of the pristine membranes are also shown in Figure 9. The FRR and the R_ir_% showed a curvature shape with respect to R_q_, where FRR showed a maximum at about 100 nm of R_q_, and similarly, R_ir_% showed a minimum at the same R_q_ value. Consequently, the general trend suggested initially an increase in FRR with the increase in R_q_. Membrane 3 does, however, show deviation from this trend. Despite its R_q_ value being the highest, it showed a lower FRR value. If more membrane had been tested with higher R_q_, the trend would have been confirmed more clearly.

The SEM images of the membranes in general look similar except for membranes 3 and 4. Membrane 3 has the highest roughness value (R_q_) among other membranes, and this is clear from both SEM and AFM images. Membranes 4 and 8 have similar R_q_ values; however, the surface morphology according to SEM images is different, with membrane 4 showing a more pressed structure compared with membrane 8. The FT-IR results indicated that all the tested membranes had a very similar chemical structure. The FT-IR results are shown in the Appendix A.

The results reported here contradict some of the experimental findings for the relationship between different types of membrane fouling and surface roughness found in the literature [25,71,74,75]. It has been reported in several works in the literature that smooth membrane surfaces are less prone to colloidal fouling compared with rough surfaces [75]. However, other workers have also observed the enhancement of anti-fouling properties and flux enhancement with rough membrane surfaces [25,76,77,78]. The increase in the FRR with the increase in R_q_ could be due to the increase in the area available for mass transfer and also creation of more turbulence that helps in removing attached particles. The further increase in R_q_ resulted in a reduction in the FRR and also increasing R_ir_%, which could be due to the presence of many valleys that can increase the fouling severity [1]. It can be said that there might be an optimum R_q_ value at which it minimizes the accumulation of the particles in the valleys and also increases the turbulence and the surface area available for mass transfer, which if exceeded may result in an adverse effect in terms of fouling. Increasing the mass transfer, and therefore the permeation rate, may eventually result in increasing the quantity of foulants approaching the surface as well.

In the work of Hobbs et al. [72], they indicated that it was the surface area difference between the image projected surface area and image surface area obtained by AFM (Á), not the average roughness of the membrane, that had a pronounced effect on the flux decline ratio. Therefore, to confirm the finding of Hobbs et al. and to check if a better correlation between FRR and the fouling percentage was obtained versus the surface area percentage difference (Á%) rather than the roughness factor R_q_, the following graphs were plotted. Appendix A shows the relationship between Á% as obtained from AFM results for each tested membrane versus the R_q_ roughness factor. The data in Appendix A can be fitted to either a straight line or an exponential equation. The best fit was found using an exponential fit, R^2^ = 0.603. When fitted to a straight-line relation the R^2^ was 0.5616. In Figure 10, the flux recovery ratio was plotted versus Á% rather than the R_q_ parameter. The data in the figure were best fitted to an exponential function as well with R^2^ = 0.566. In this figure, the SW30-XLE (membrane 3) did not show an off behavior as in Figure 7 and Figure 8. This might be an indication that it was not the absolute value of the R_q_ parameter that had a clear effect on the fouling and flux recovery ratio; however, it was Á% that had a more pronounced effect on the amount of fouling on the membrane. Therefore, the Á% parameter was a better representation of the membrane surface roughness compared with the R_q_ parameter. Membranes 6 and 7 were both somewhat off trend; however, this did not result in changing the overall fit, which was the increase in FRR with the degree of wrinkling of the surface of the membrane represented in Á%.

Similar results were obtained (Figure 11) for the R_ir_% versus the Á%. The data in Figure 11 were best fitted to a logarithmic equation with R^2^ = 0.608; they could be fitted to a straight-line equation as well, however, with lower R^2^ = 0.536. There was a clear reduction in the irreversible fouling percentage with an increasing surface area difference of the membrane, and again, membrane 3 did not show off results in this case.

The roughness of the surface was measured for both the pristine and fouled membranes (Figure 12). For all membranes except for membrane 2, the R_q_ decreased after fouling. The decrease in R_q_ for membrane 1 was the lowest, as shown in Figure 12. The decrease in the roughness of the membranes after fouling could be explained by the fouling layer filling surface depressions and making the surface smoother. Both membranes 1 and 2 had the lowest R_q_ values among other membranes, R_q_ = 48.6 and 75.8 nm, respectively (unfouled membranes). This may explain the increase in the surface roughness of membrane 2 as the surface was already smooth, and the deposition of foulants added more roughness to the surface, or the increase in roughness for the case of membrane 2 could be attributed to the uneven coverage of particles on relatively smooth membrane surface [79]. To demonstrate the effect of Á% rather than the fouling parameter R_q_ only, Figure 13 shows the Á% for pristine and fouled membranes for the set of eight membranes tested here.

All membranes showed a decrease in the Á% after fouling (Figure 13). This would suggest that Á% was a better indication of the surface roughness of the membrane, compared with the R_q_ value. Also, a very clear correlation (R^2^ = 0.72) was found between both FRR and the R_ir_% with the change in the surface area percentage difference (ΔÁ%) between the pristine and the fouled membrane as shown in Appendix A, respectively. This confirmed that the amount of fouling on the membrane decreased with the surface roughness, which would be better expressed in terms of Á% rather than only R_q_. As R_q_ was the root mean squared deviation of height data within an AFM height profile, it was easily affected by small numbers of pixels within the image showing large variations, such as by membrane surface defects, with the surface area difference less prone to these effects [80]. This made measures of the variation in surface area for a scanned plane, as embodied by the surface area difference and Wenzel’s roughness factor rather than height deviation used for R_q_ and average roughness, potentially more reliable as a measure of the surface roughness where changes to interaction or interfacial area with increasing roughness are important [81].

### 4.3. Correlations Between Water Contact Angle, Fouling Percentage and FRR

It has been observed in the literature [1,22,82] that in general, the greater the hydrophilicity of the membrane surface, the lower the fouling tendency of the membrane. Explanations usually assume that a thin layer of bonded water exists on the surface of the hydrophilic membrane due to the formation of the hydrogen bonding, acting as an effective barrier to prevent or reduce undesirable membrane fouling [83]. Also, the higher hydrophilicity of the membrane is often expected to result in higher permeation flow rates through lowered resistance, which may also increase the rate of approach of foulants to the surface of the membrane. Yin et al. [28] reported that membranes with moderate roughness and the most hydrophobic surface had the highest fouling reversibility, and this was mainly due to the hydrophilicity of the foulant present in the type of waste water tested. Several studies have shown enhancement in the pure water flux with the increase in hydrophilicity of the membrane [1,28,84,85,86]. However, a clear trend between the water contact angle and the amount of fouling for a set number of membranes or for a quite wide range of contact angle values is rarely shown. Bildyukevich et al. [83] indicated that the protein adsorption to six industrial membrane surfaces showed no correlation with the membrane hydrophilicity. To further investigate the effect of membrane hydrophilicity on the flux recovery ratio and the amount of fouling, the water contact angles for all of the membranes were measured as described previously, and the average water contact angle values are presented in Table 3.

Figure 14, Figure 15, and Appendix A show the FRR, R_ir_%, and R_t_%, respectively, as a function of the water contact angle and θ values of the studied commercial membranes. All three figures did not show any clear correlation (linear fitting gives R^2^ = 0.09) between θ and either FRR, R_ir_%, or R_t_%. This indicates that the hydrophilicity of the membrane is not a major factor affecting either the flux recovery ratio or the fouling percentage. The roughness expressed in Á% and the HSP distance presented in previous sections showed more clear trends and correlation compared with the hydrophilicity of the membrane; however, most of the membrane fabrications and modifications in the literature are based on the enhancement of the hydrophilicity of the membrane for higher permeation flux and lower fouling.

As was previously shown in Figure 2, membrane 4 showed the highest permeation rate; however, this membrane was the most hydrophobic membrane with θ = 94.62°. Despite the high permeation flow rate of this membrane, its permeation reduction percentage was neither the lowest nor the highest, as indicated in Figure 3. The two membranes with the highest hydrophilicities were membranes 2 and 3 with θ values of 39.46 and 39.51°, respectively, and showed the lowest permeation flow rates. They also showed very similar percentage permeation reduction as shown in Figure 3 (above 20%). In Figure 14 and Figure 15, membrane 1 consistently deviates from the trend. This membrane (UTC73) showed the highest fouling percentage and the lowest flux recovery ratio, although it was not the most hydrophobic membrane. Membrane 1 had the lowest FRR and the lowest Ra and Á% values among all tested membranes, which may explain its lowest FRR.

### 4.4. Estimating the FRR from Á%, Ra, and θ

The fouling rate likely resulted from a complex combination of contributing factors, which should be considered together, rather than in isolation. For instance, membranes 6 and 7 had similar FRR values and similar Á% values (Figure 1 and Figure 10), 12.69 and 12.75, respectively, although in terms of Ra (Figure 7) and water contact angle (Figure 14), they were markedly different. This underlines how using a single factor to understand reversible fouling is insufficient.

From our results, both the HSP distance and the roughness of the membranes expressed in the Á% showed a good correlation with either the FRR or the R_ir_%, indicating that they must play a role in the attachment of BSA molecules to the membrane surface. The water contact angle of the membranes did not show any clear correlation. To understand the contribution of each of those surface properties on the flux recovery ratio, the data were fitted to two empirical models. The experimental data that were used to fit the models are represented in Table 3.

We propose two empirical models, assuming that the Ra, Á%, and θ were the independent variables and the dependent variable (output) was the FRR. The two proposed models are as shown in Equations (6) and (7).
(6)$$\mathrm{FRR}=a{\left(A^{\prime }\%\right)}^{n}{\mathrm{Ra}}^{m}\theta^{k}$$
(7)$$\mathrm{FRR}=a{\left(A^{\prime }\%\right)}^{n}+{\mathrm{bRa}}^{m}+{c\theta }^{k}$$

The fitting coefficients, a, b, n, m, c, and k, were estimated using the Solver function in Excel via minimizing the errors between the experimental FRR and the calculated FRR (least square method). It was assumed that all three variables, Á%, Ra, and θ, were directly proportional to the FRR, and by optimizing the fitting parameters, the dependency of the output on those three variables would be clearer as the signs of the exponents n, m, and k would adjust for any inverse proportionality with the FRR. The results of the fitting parameters for both models are shown in Table 4 with the R^2^ and the root mean square error (RMSE) of the optimized solution. Those fitting criteria of R^2^ and RMSE were the best possible for these data. Both models showed very similar fitting criteria, and there was no clear advantage of either. However, the fitting parameters can give an indication on the weight and the effect of each parameter on the FRR. For example, with model 1, from the values of the exponents n, m, and k, it showed that θ had the lowest influence on the FRR as its exponent k = 0.06, while the Ra and Á% showed a higher exponent as indicated in the values of parameters n and m, as shown in Table 4. Model 2 had three extra weighing parameters for each input parameter, a, b, and c. From the values of the weights, it was clear that the highest weight corresponded to the Á%. The weights of Ra and θ were 1.15 and 1.37, respectively; however, the exponent m for the Ra variable was much greater than the exponent k for θ variable (1.5 and 0.41, respectively). This in turn led to the same conclusion as model 1, stating that Á% and Ra had a higher contribution to the flux recovery ratio compared with θ (hydrophilicity).

Figure 16 shows the average percentage contribution (for the eight commercial membranes) of each variable in both models 1 and 2 to the FRR based on the fitting parameter estimated in this work. Both models showed that the highest contribution to the FRR was due to the Á%, followed by Ra and finally the hydrophilicity of the membranes, θ. The Á% showed about 62% contribution to the FRR based on model 2; however, it showed about 39% contribution based on model 1. The Ra had a contribution of about 37% for model 1 and 29.3% for model 2. The hydrophilicity contribution represented in θ was found to be only 8.8% for model 2 and 23.8% for model 1.

We believe that the FRR depended on more surface property variables that were not all included in both models. The purpose of fitting those models was to understand the weight of each of the different surface properties chosen here on the flux recovery ratio and therefore on the fouling characteristics of the membrane. The main conclusion drawn here was that it was not the hydrophilicity of the membrane that had the major effect on the amount of fouling and its reversibly; it was the surface area difference of the membrane and the HSP distance between the foulant molecule and the membrane surface (affinity) that had the major effect on the amount of fouling and its reversibility.

### 4.5. The Selection of Proper Materials Using HSP for the Synthesis of Anti-Fouling Materials

Based on these results, and since the HSP distance contributed to about a third of the FRR, polymeric materials with potentially lowered fouling rates could be selected based on the HSP distance to the foulant rather than focusing only on the hydrophilicity of the membrane. For example, the ternary diagrams shown in Figure 17 and Figure 18 show the distances between BSA molecules and various commercial membranes (Figure 17) and other randomly chosen polymers from the HSPiP database (Figure 18). Figure 17 shows the eight tested commercial membranes and other additional commercial membranes, whose HSP values were measured using the method described in Section 3.5. Most of those commercial membranes were located in the same space, and therefore, their HSP distances to BSA were quite similar. All of those commercial membranes were made of polyamide, which made their affinities to BSA quite similar. Figure 18 shows that there are several polymers that are located far away from BSA molecule. It is also observed that all polymers are located at the lower right corner. No polymers are located at the top or left corner. Using Figure 18, one may be able to make a preliminary estimation of possible polymers to be tested for the synthesis of a superior antifouling material. If a polymer is located far away from the BSA molecule in the ternary diagram, it does not mean that it can be used as a membrane material, as the mechanical properties of this polymer is also an important factor to keep in mind. However, this method can be used as a preliminary screening step of the various polymers that show very low affinity to BSA or to any other foulant molecule understudy assuming its HSP is known or can be measured.

## 5. Conclusions

Eight commercial membranes were tested for their BSA protein fouling during dead-end filtration experiments. The effect of the HSP distance between the BSA molecule and the membrane surfaces, surface roughness, surface area percentage difference, and hydrophilicity of the membranes on the flux recovery ratio, and therefore the irreversible fouling percentage, was presented and discussed. The results indicated that the hydrophilicity of the membranes was not a major factor affecting the amount of fouling and the flux recovery ratio of the membranes. Rather it was the surface area percentage difference Á% and the affinity represented in the HSP distance between the membrane surface and the foulant molecule that had the major impact on the reversibility of foulants and the ability to recover the flux by backwashing. Also, the surface roughness of the membrane was better represented by Á% as the R_q_ value alone may give a misleading result regarding the surface roughness. Therefore, it is believed that Á% is a better representation of how wrinkled the membrane surface is. The results showed that the more wrinkled the surface is, the better the flux recovery and the lower the irreversible fouling percentage, likely due to higher turbulence close to the selective layer for the more wrinkled membrane surface. According to the HSP results shown here, the surfaces that had a higher value of the hydrogen bonding components (close to the δ_h_ of the BSA molecule) seemed to be more prone to fouling, suggesting that BSA molecules became attached to the surface irreversibly via hydrogen bonding with the membrane surface. Therefore, we recommend when selecting a material for membrane fabrication, its HSP compared with the HSP of the possible foulant present in the solution should be taken in consideration for lower fouling problems and therefore better membrane performance.

## Figures and Tables

**Figure 1 membranes-14-00235-f001:**
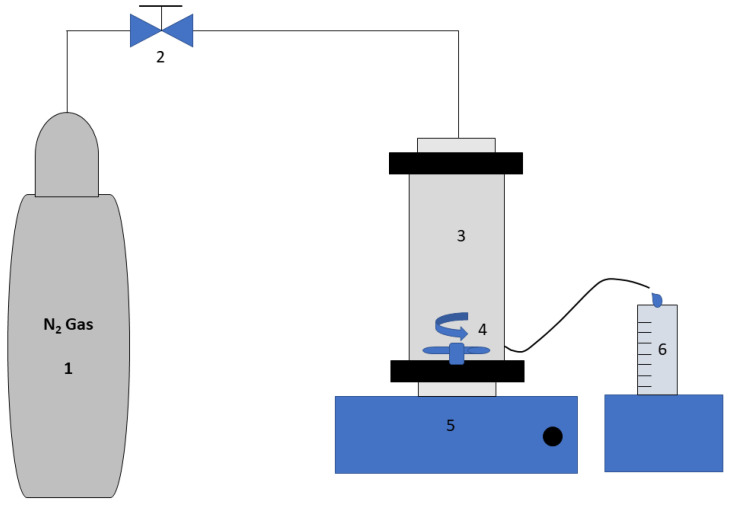
The dead-end filtration setup: 1, N_2_ gas cylinder; 2, valve; 3, dead-end stirred cell; 4, magnetic stirrer; 5, magnetic stirrer plate; 6, graduated cylinder.

**Figure 2 membranes-14-00235-f002:**
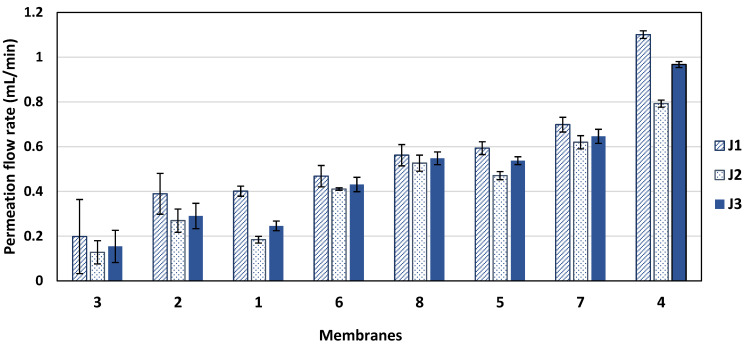
The permeation flow rates before, during, and after fouling. J_1_, steady state permeation flow rate for pure water; J_2_, steady state permeation flow rate for BSA solution; and J_3_, steady state permeation flow rate for pure water after backwashing the membrane. Error bars represent standard error. Membranes are numbered as given in Table 1 and ordered by average permeation rate.

**Figure 3 membranes-14-00235-f003:**
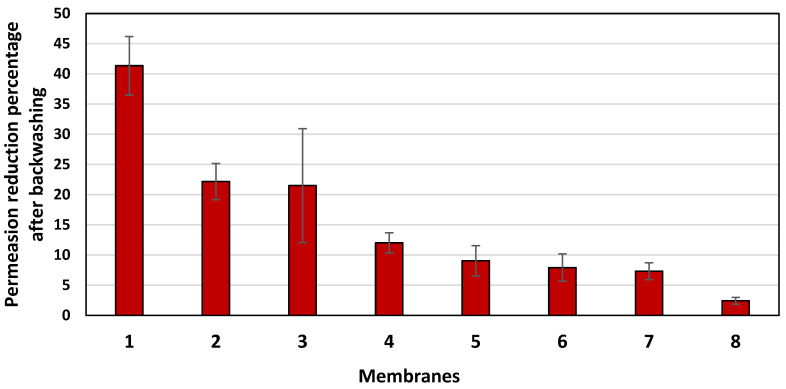
The percentage permeation rate reduction after backwashing for each of the eight commercial membranes run under the same experimental conditions. Error bars show the standard error.

**Figure 4 membranes-14-00235-f004:**
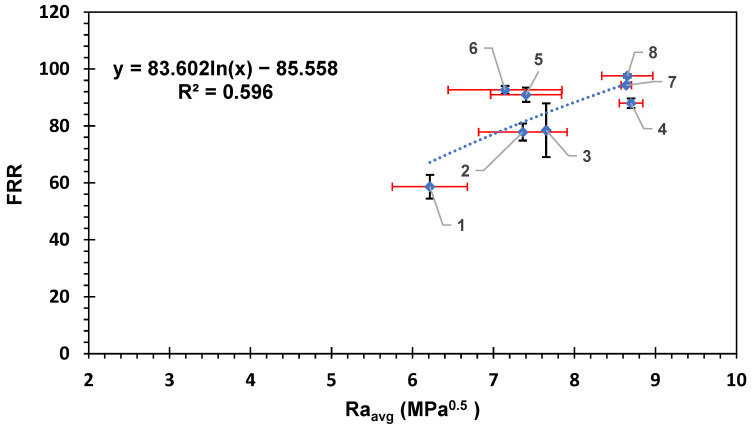
The flux recovery ratio versus the average HSP distance to the BSA molecule for each commercial membrane tested. The error bars represent the standard error for both FRR and Ra_avg_. Red error bars show the standard error in Ra_avg_ while black error bars show the standard error in FRR. Each membrane is numbered according to the scheme in Table 2.

**Figure 5 membranes-14-00235-f005:**
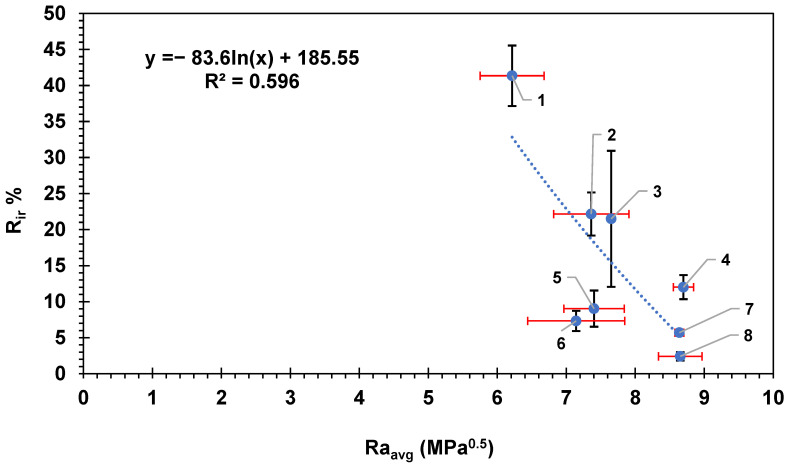
The irreversible fouling, R_ir_%, versus the average HSP distance to BSA, Ra_avg_, for the eight commercial membranes. The error bars represent the standard error. Red error bars show the standard error in Ra_avg_ while black error bars show the standard error in (R_ir_%). Each membrane is numbered according to the scheme in Table 2.

**Figure 6 membranes-14-00235-f006:**
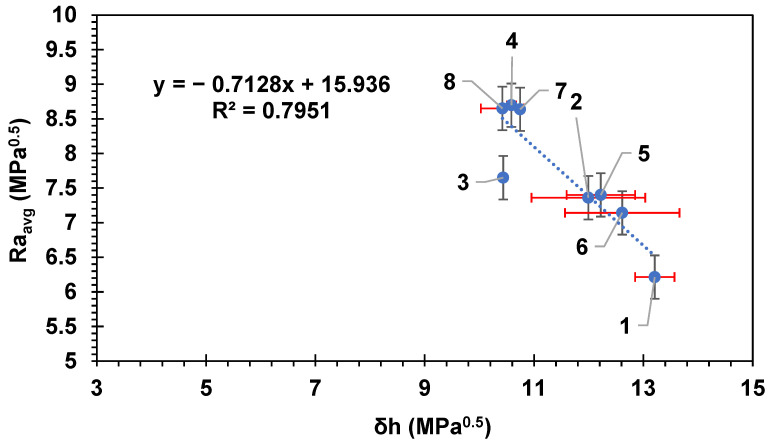
The average HSP distance between BSA, Ra_avg_, and each commercial membrane versus the hydrogen bonding component δ_H_ of each membrane. Error bars represent the standard error. Red error bars show the standard error in δh while black error bars show the standard error in Ra_avg_. Each membrane is numbered according to the scheme in Table 2.

**Figure 7 membranes-14-00235-f007:**
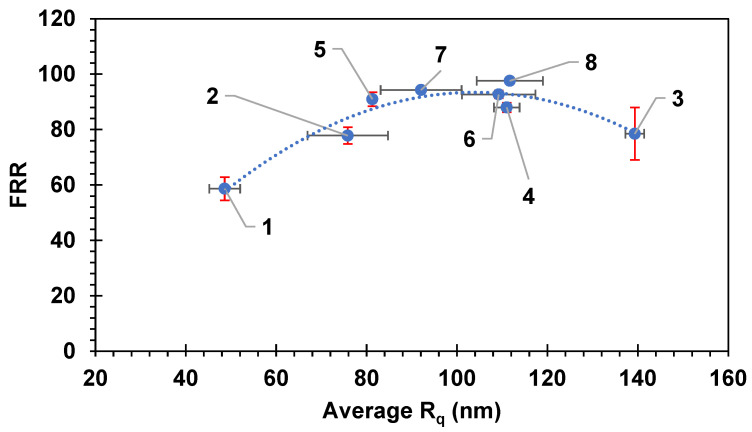
The flux recovery ratio versus the average roughness (R_q_). Each membrane is numbered according to the scheme in Table 2. Error bars represent the standard error. Red error bars show the standard error in FRR while black error bars show the standard error in Average R_q_.

**Figure 8 membranes-14-00235-f008:**
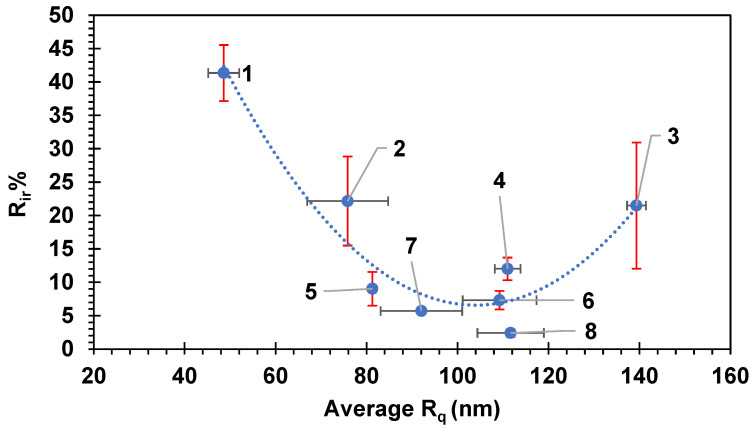
The irreversible fouling percentage versus the average membrane roughness (R_q_). Each membrane is numbered according to the scheme in Table 2. Error bars show the standard error. Red error bars show the standard error in R_ir_% while black error bars show the standard error in R_q_.

**Figure 9 membranes-14-00235-f009:**
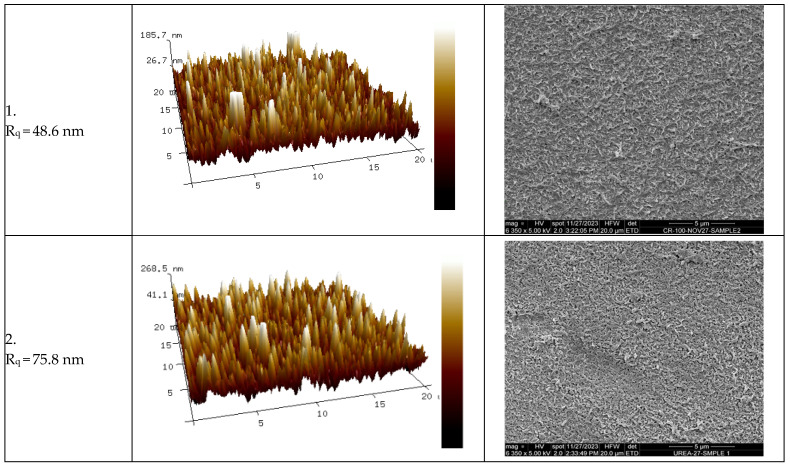
AFM 3D images (20 μm scan size) of the eight commercial membranes with their R_q_ surface roughness values and SEM images of the surface of the membranes (20 μm). Each membrane is numbered according to the scheme in Table 2.

**Figure 10 membranes-14-00235-f010:**
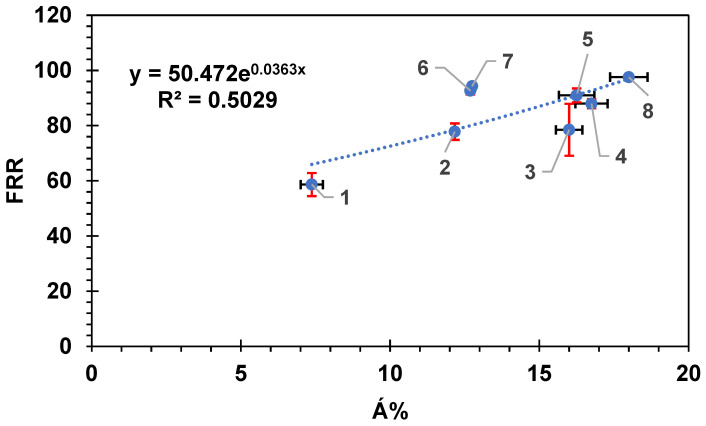
The flux recovery ratio versus the surface area percentage difference (Á%) for the eight commercial membranes. Each membrane is numbered according to the scheme in Table 2. Error bars show the standard error. Red error bars show the standard error in FRR while black error bars show the standard error in (Á%).

**Figure 11 membranes-14-00235-f011:**
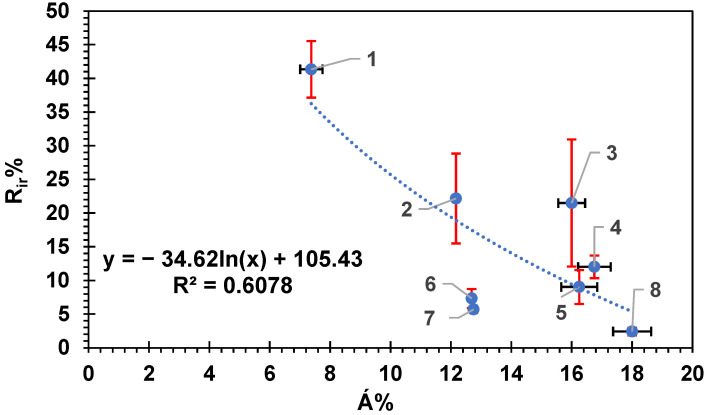
The irreversible fouling percentage R_ir_% versus the surface area percentage difference (Á%) for all tested membranes. Each membrane is numbered according to the scheme in Table 2. Error bars show the standard error. Red error bars show the standard error in (R_ir_%) while black error bars show the standard error in (Á%).

**Figure 12 membranes-14-00235-f012:**
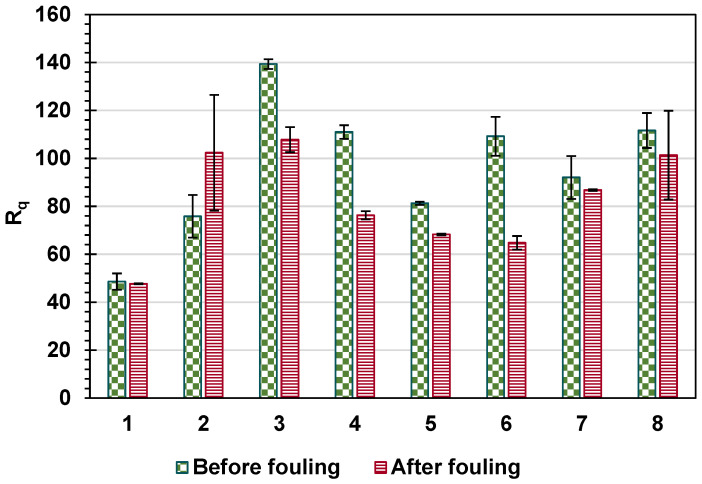
The average surface roughness (R_q_) before and after fouling. Error bars show the standard error.

**Figure 13 membranes-14-00235-f013:**
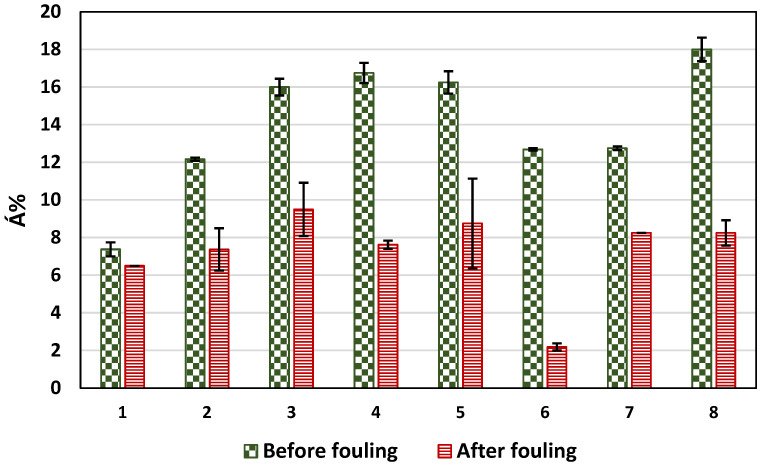
The average surface area difference before and after fouling. Error bars show the standard error.

**Figure 14 membranes-14-00235-f014:**
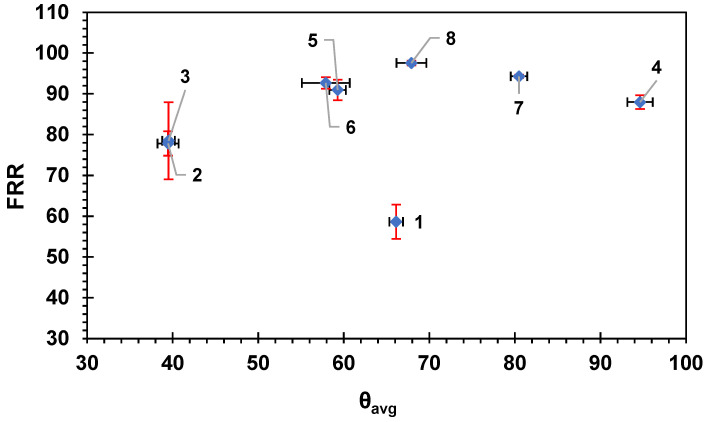
The flux recovery ratio versus the average water contact angle. Each membrane is numbered according to the scheme in Table 2. Error bars show the standard error. Red error bars show the standard error in FRR while black error bars show the standard error in θ_avg_.

**Figure 15 membranes-14-00235-f015:**
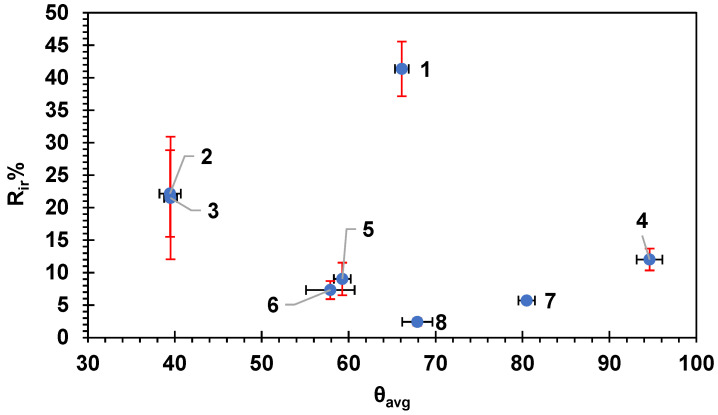
The irreversible fouling percentage R_ir_% versus the average water contact angle (θ_avg_) on the membranes. Each membrane is numbered according to the scheme in Table 2. Error bars show the standard error. Red error bars show the standard error in R_ir_% while black error bars show the standard error in θ_avg_.

**Figure 16 membranes-14-00235-f016:**
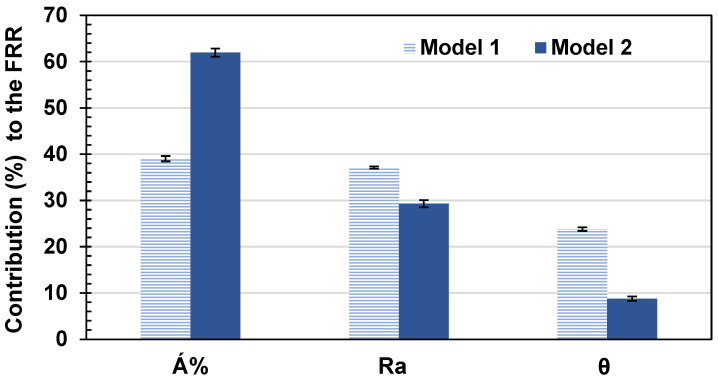
The percentage contribution of each of the input variable to FRR. Error bars show the standard error.

**Figure 17 membranes-14-00235-f017:**
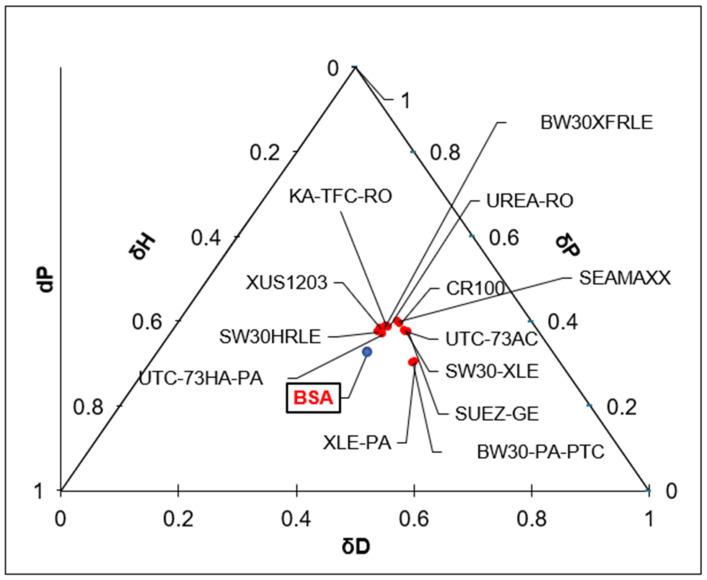
The ternary diagram showing the HSP values of the BSA protein in comparison with the HSP of a list of commercial membranes.

**Figure 18 membranes-14-00235-f018:**
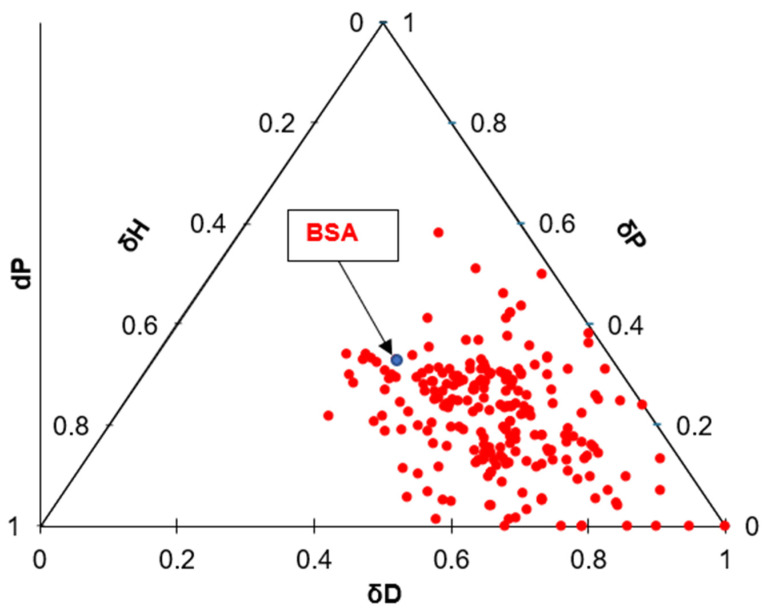
The ternary diagram showing the HSP values of the BSA protein in comparison with the HSP of a list of polymers from Hansen Data base. Red dots show the location of various polymers in the ternary diagrams, with BSA shown in blue.

**Table 1 membranes-14-00235-t001:** RO STERLITECH commercial membranes.

Number	Commercial Membrane
1	UTC-73HA-PA-RO
2	X201, PA-UREA, RO
3	SW30XLE-PA-TFC-RO
4	XLE-PA-TFC-RO
5	AK-TFC RO
6	BW30XFRLE
7	BW30-PA-TFC-RO
8	CR100

**Table 2 membranes-14-00235-t002:** The estimated HSP values of the commercial membranes and BSA protein, with the calculated HSP distance Ra for each.

Units: Mpa^0.5^	δ_D_	δ_P_	δ_H_	δ_T_	Ra_avg_- to BSA
Commercial Membranes					
1	17.88 ± 0.54	18.11 ± 0.95	13.21 ± 0.36	28.72 ± 0.36	6.21 ± 0.46
2	17.58 ± 0.22	18.53 ± 0.39	11.99 ± 1.04	28.25 ± 0.54	7.36 ± 0.55
3	18.51 ± 0.06	17.29 ± 0.08	10.43 ± 0.02	27.39 ± 0.02	7.65 ± 0.02
4	19.31 ± 0.05	13.06 ± 0.18	10.58 ± 0.08	25.61 ± 0.13	8.69 ± 0.15
5	17.38 ± 0.02	18.86 ± 0.38	12.22 ± 0.63	28.43 ± 0.44	7.40 ± 0.44
6	17.37 ± 0.03	18.61 ± 0.22	12.61 ± 1.05	28.44 ± 0.53	7.14 ± 0.70
7	19.25 ± 0.08	12.98 ± 0.11	10.74 ± 0.04	25.58 ± 0.09	8.64 ± 0.06
8	17.43 ± 0.01	18.16 ± 0.27	10.42 ± 0.39	27.24 ± 0.32	8.65 ± 0.32
BSA protein	19.9	18.2	17.5	32.15	0

**Table 3 membranes-14-00235-t003:** The experimental data of flux recovery ratio, surface area percentage difference, HSP distance, and water contact angle for each commercial membrane tested.

Membrane	FRR	Ra (Mpa^0.5^)	Á%	θ°
UTC-73HA-PA-RO	58.65	6.21	7.38	66.11
X201, PA-UREA, RO	77.83	7.36	12.17	39.46
SW30XLE-PA-TFC-RO	78.50	7.65	16.00	39.51
XLE-PA-TFC-RO	87.98	8.69	16.75	94.62
AK-TFC RO	90.96	7.40	16.25	59.27
BW30XFRLE	92.67	7.14	12.69	57.89
BW30-PA-TFC-RO	94.29	8.64	12.75	80.47
CR100	97.58	8.65	18.00	67.90

**Table 4 membranes-14-00235-t004:** The two proposed models and their estimated optimized parameters for the estimation of the FRR as a function of Á%, Ra, and θ.

No.	Model	Fitting Parameters							
		a	b	c	n	m	k	R^2^	RMSE
1	$FRR=a{\left(A^{\prime }\%\right)}^{n}Ra^{m}\theta^{k}$	14.63	-	-	0.29	0.35	0.06	0.66	6.93
2	$FRR=a{\left(A^{\prime }\%\right)}^{n}+bRa^{m}+c\theta^{k}$	16.56	1.15	1.37	0.44	1.50	0.41	0.66	6.98

## Data Availability

The original contributions presented in the study are included in the article/Appendix A; further inquiries can be directed to the corresponding author.

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
