# Peer review of "Combined Effects of Surface Roughness, Solubility Parameters, and Hydrophilicity on Biofouling of Reverse Osmosis Membranes"

_membranes, 2024, doi:10.3390/membranes14110235_

Round 1

Reviewer 1 Report

Comments and Suggestions for Authors

This study investigates the impact of the surface properties of TFC membranes on membrane fouling. The main influencing factors are not the hydrophilicity of the membrane surface, but rather its surface roughness and the HSP distance. The paper meets the requirements of the journal Membranes, but there are the following issues:

  1. The abstract is too long; it is recommended to shorten it.
  2. There are too many figures in the paper. Some figures could be combined into one large figure or moved to the Supplementary Information.
  3. Why did you choose BSA? If you used other foulants, such as polysaccharides, would the conclusions be different?
  4. It is suggested to provide a table summarizing the parameters and surface properties of the commercial membranes tested.

Author Response

Reviewer 1: This study investigates the impact of the surface properties of TFC membranes on membrane fouling. The main influencing factors are not the hydrophilicity of the membrane surface, but rather its surface roughness and the HSP distance. The paper meets the requirements of the journal Membranes, but there are the following issues:

  1. The abstract is too long; it is recommended to shorten it.

Abstract has been shortened in accordance with the reviewer recommendations.

  1. There are too many figures in the paper. Some figures could be combined into one large figure or moved to the Supplementary Information.

We have reduced the number of figures by moving several to a supplementary materials file.

  1. Why did you choose BSA? If you used other foulants, such as polysaccharides, would the conclusions be different?
  2.  

BSA was chosen as a model protein, which is one component of the biofilms that are attached to the membrane surface. BSA was chosen as a model protein due to the use in other studies of fouling, allowing comparisons to be made. Certainly differences in the properties of the other biomacromolecules will affect surface adsorption.

  1. It is suggested to provide a table summarizing the parameters and surface properties of the commercial membranes tested.

A table containing the specifications of each of those commercial membranes has been added in the supplementary document as well.

Reviewer 2 Report

Comments and Suggestions for Authors

Neveen AlQasas and Daniel Johnson investigated the cause of bio-fouling of BSA in 8 various commercial membranes and then related the reversible and irreversible flux with some membrane properties. They further concluded that the Hansen solubility parameter differences between foulant and membrane and the surface area difference between the image projected surface area and image surface area play more important roles than the hydrophilicity and roughness. Both the topic and conclusion are pretty interesting and also crucial to the membrane field. I would suggest it be published after the following issues are addressed.

1. In a RO test, the fouling-induced flux loss is typically observed as a continuing decreased flux with time. In the current work, a dead-end RO test was employed. When the authors referred to flux, which flux is exactly chosen, overall flux, the steady flux at a specified period, or others?

2.       In the BSA permeation test, how long the tests were performed?

3.       Different commercial RO membranes are designed for various conditions and thus their performances will and also should behave differently. I would suggest the authors list more info about the 8 membranes used in this work, including but not limited to the flux, salt rejection, tested condition provided by the manufacturer, the active layer’s materials, etc.

4.       Is 8 bars sufficient for an RO test? Some RO membranes are designed for a much higher operation condition, like 30 bar. At a transmembrane pressure much lower than it, the membrane may not behave as it should.

5.       In Figures 2 and 3, the names of membranes were used rather than the membrane code. Would authors switch them to the membrane code?  

6.       In the SEM and AFM tests, the pristine membranes were used. Considering all RO membranes include additives and hence need to be pre-treated for use, the membranes after 1st water run would be a better choice for morphology study.

7.       Using A% rather than roughness indeed solves the off-trend issue of membrane 3, but membranes 6 and 7 are off-trend now. Any explanation?

8.       Since the R2s are very low in all fitting reported in this work, I see few points to report them in this work, as well as section 4.4, for that using 6 parameters to fit a database with 8 data does not provide a lot of meaning.

9.       What do the blue dots in Figures 21 and 22 stand for?

Author Response

Reviewer 2:

Neveen AlQasas and Daniel Johnson investigated the cause of bio-fouling of BSA in 8 various commercial membranes and then related the reversible and irreversible flux with some membrane properties. They further concluded that the Hansen solubility parameter differences between foulant and membrane and the surface area difference between the image projected surface area and image surface area play more important roles than the hydrophilicity and roughness. Both the topic and conclusion are pretty interesting and also crucial to the membrane field. I would suggest it be published after the following issues are addressed.

  1. In a RO test, the fouling-induced flux loss is typically observed as a continuing decreased flux with time. In the current work, a dead-end RO test was employed. When the authors referred to flux, which flux is exactly chosen, overall flux, the steady flux at a specified period, or others?

Each filtration test was run for 2 hours, and the volume of permeate was recorded with time and therefore the permeation flow rate was calculated. The permeation rate was almost constant after approximately 30 minutes from the starting of the experiment, the steady state flow rate was considered as the average flow rate for the duration of the experiment after stabilization (the average flow rate for the last hour of experiment). We have updated the manuscript to make this information clearer.

In the BSA permeation test, how long the tests were performed?

Filtration test were carried out for two hours. Manuscript has been amended as per previous reply to make this clearer,.

  1. Different commercial RO membranes are designed for various conditions and thus their performances will and also should behave differently. I would suggest the authors list more info about the 8 membranes used in this work, including but not limited to the flux, salt rejection, tested condition provided by the manufacturer, the active layer’s materials, etc.

We have added a table detailing manufacturer’s specifications for each membrane as part of the supplementary materials.

  1. Is 8 bars sufficient for an RO test? Some RO membranes are designed for a much higher operation condition, like 30 bar. At a transmembrane pressure much lower than it, the membrane may not behave as it should.

Whilst all of the membranes used in these tests are RO membranes, they are mainly manufactured for brackish water or industrial wastewater treatment. The operating pressure of 8 bars is thus within the normal operating range for these membranes.

  1. In Figures 2 and 3, the names of membranes were used rather than the membrane code. Would authors switch them to the membrane code?  

Figures 2 and 3 have been modified in the manuscript by replacing the names of the membranes with the numbers indicated, and in the body of the manuscript as well.

  1. In the SEM and AFM tests, the pristine membranes were used. Considering all RO membranes include additives and hence need to be pre-treated for use, the membranes after 1stwater run would be a better choice for morphology study.

Pristine membranes were soaked in deionized water for a minimum of two days to remove soluble additives.

  1. Using A% rather than roughness indeed solves the off-trend issue of membrane 3, but membranes 6 and 7 are off-trend now. Any explanation?

As shown in figures 10 and 11, both membranes 6 and 7 showed a small deviation from the trend, which resulted in lowering the R2, however we did not consider as outliers because they did not change the overall trend. Whilst this study capture the generally accepted major contributors to fouling (hydrophilicity, surface roughness and polar and dispersion nature of foulant and membrane), there may be other contributing factors.

Since the tested membranes are commercial membranes, and when studying their surface properties in terms of their effect on the FRR, we are not isolating one property while changing another, therefore the influence of one surface property may be more prominent in certain cases. As for the case of those two membranes, most likely the increase in FRR for membrane 7 compared to membrane 6 is not due to the increase in the Á%, it may be a result of the increase in Ra, or the reduced affinity toward BSA molecule for membrane 7 compared to membrane 6.

 A paragraph was added in the manuscript to expand on this point.  

  1. Since the R2s are very low in all fitting reported in this work, I see few points to report them in this work, as well as section 4.4, for that using 6 parameters to fit a database with 8 data does not provide a lot of meaning.

The models did not demonstrate the best fit, as indicated by the low R² values. However, the goal of fitting these models was not to propose a new method for estimating FRR, but rather to understand the relative influence and impact of each parameter on membrane fouling. As noted in the manuscript, membrane fouling is influenced by a large range of surface properties and factors. This study attempted to capture the relative contribution of several factors generally accepted to be major contributors and also assessed the utility of using HSP data toad to our understanding of membrane fouling. It is expected that including more surface properties would have improved the model’s accuracy.

 Regarding the number of data points (8), while this is insufficient for fitting a highly accurate model, only eight commercial membranes were studied. Given that the purpose of these models was as mentioned, the use of these eight data points was appropriate for the time being. Collecting more data points would require testing additional commercial membranes and conducting more experiments, which would extend the timeline, as each experiment was repeated multiple times to ensure accuracy and reliability. Also, the two proposed empirical models showed the same accuracy and gave the same conclusion, for example model 1, which has a lower fitting parameter (4 fitting parameters) showed the same accuracy as the model 2 with more fitting parameters (6 fitting parameters). The main conclusion here is that both of them are indicating that the surface roughness represented in A’% and the affinity represented in Ra have a more pronounce effect on membrane fouling compared to hydrophilicity represented by the water contact angle.

  1. What do the blue dots in Figures 21 and 22 stand for?

The blue represent the location of the HSP values of BSA in the ternary diagram, as labelled.